# FLAT IS THE NEW SHARP: FLATNESS-AWARE REGULARIZATION FOR ROBUST LEARNING

## ABSTRACT

Understanding and improving the generalization of neural networks has been a central focus in machine learning. One of the most significant efforts to address this challenge revolves around the concept of the loss landscape in deep neural networks (DNNs). While some researchers have posited that solutions located in flatter regions of the loss surface tend to generalize better than those in sharper regions, others have provided theoretical frameworks and empirical findings suggesting that flat minima are not the sole, or even primary, reason for strong generalization. Despite these advances, the relationship between loss landscape geometry and generalization remains an open question. In this work, we contribute to this open question by introducing Flatness-Aware Regularization (FA-Regularization). This method explicitly penalizes the loss surface towards flatter minima by incorporating an estimate of the trace of the squared Hessian into the training loss. We present empirical results demonstrating that this Hessian estimate effectively penalizes the curvature of the loss surface, enabling the optimizer to converge to flatter regions. We tested our FA-Regularizer across a variety of models (MLP and Logistic Regression) and datasets (CIFAR-100, IMDB Movie Reviews, and Breast Cancer Wisconsin). Our FA-Regularization method consistently leads to improved generalization on cifar-100 compared to a baseline loss function without the penalty term. Our FA-Regularization method indicates that, flatness is shown to correlate with, but not fully explain, generalization..

## 1 INTRODUCTION

Generalization in deep neural networks remains a fundamental challenge, and understanding the geometry of the loss landscape has become central to addressing this challenge. It has been suggested (Hochreiter & Schmidhuber, 1997) that solutions lying in flatter regions of the loss surface tend to generalize better than those in sharper regions– a view further supported by evidence showing that training with small mini-batches often converges to flatter minima and yields improved generalization (Keskar et al., 2017). Despite these findings, the precise role of loss landscape geometry in generalization remains unresolved. Some theoretical work (Dinh et al., 2017) have also shown that sharp minima can even generalize well, indicating that flatness alone is not a complete explanation. Nonetheless, the prevailing consensus is that flatter minima are generally associated with stronger generalization, while sharper minima often correspond to overfitting and poor performance on unseen data.

In light of these insights and in a bid to provide answers to the open problems around the relationship between the loss-landscape and generalization, we propose *Flatness-Aware Regularization (FA-Regularization)*, a principled approach to promote flat loss landscapes during training. The core idea is to explicitly penalize the curvature of the training loss, encouraging the optimizer to find flatter regions of the landscape. We do this for 2 reasons: (1) to contribute to the ongoing discussion among the academic community, with the hope of providing empirical clarity on the role of flatness in generalization, and (2) to evaluate whether explicitly enforcing flatness can improve stability and predictive performance across diverse learning settings. Concretely, FA-Regularization introduces an additional term into the training objective that estimates the overall curvature via the trace of the squared Hessian matrix. By penalizing this curvature measure, the optimization process is guided towards a minima with lower sharpness (i.e., flatter minima), which we hypothesize will improve the model's generalization ability. We implement this by using an efficient stochastic approximation to

compute the trace of the Hessian, so that the curvature penalty is tractable even in high-dimensional parameter spaces. Importantly, our approach seamlessly integrates into standard training pipelines: it works out-of-the-box with popular optimizers like SGD and Adam, without requiring any special optimization procedures or custom solvers.

By effectively flattening the loss landscape through this regularizer, we aim to improve generalization performance. We verify via visualizations of the loss surface that adding the flatness-aware penalty leads the optimizer to converge at a flatter minimum than standard training. We then empirically evaluate the impact of FA-Regularization on model generalization. In our experiments, we train neural networks on benchmark tasks with and without the flatness-aware penalty and compare their performance on unseen test data. This evaluation allows us to assess whether enforcing a flatter loss landscape during training indeed translates into improved generalization performance relative to the conventional training without FA-Regularization.

Our contributions are as follows:

- **Flatness-Promoting Regularizer:** We introduce a novel flatness-aware regularization technique that explicitly penalizes the curvature of the loss surface. This approach is general and can be applied alongside any standard differentiable loss function to encourage the optimizer to find flatter minima.
- **Compatibility with Standard Optimizers:** The proposed FA-Regularization seamlessly integrates into existing training pipelines. It requires no special optimization procedure; for example, it works out-of-the-box with popular optimizers like SGD and Adam, without the need for auxiliary adversarial loops or custom solvers.
- **Improved Generalization Performance:** Through extensive experiments, we demonstrate that models trained with FA-Regularization consistently exhibit better generalization on unseen data compared to models trained without this regularizer.

Section 2 below develops the framework and theoretical foundation of our proposed *Flatness-Aware Regularization (FA-Regularization)*, including the definition of a flatness measure, efficient estimation via Hutchinson's method, and the resulting regularized training algorithm. Section 3 presents our experimental evaluation, where we assess the effectiveness of FA-Regularization across multiple datasets and architectures. Section 4 gives a detailed review of related works that motivate and contextualize our study. Finally, Section 5 concludes with a discussion of findings, limitations, and directions for future research.

## 2 FLATNESS-AWARE REGULARIZATION (FA-REGULARIZATION)

This section develops the framework and theoretical foundation behind our FA-Regularization. Our central motivation here is to explicitly encourage the optimization trajectory to converge towards flatter regions of the loss landscape.

### 2.1 MEASURE OF LOCAL FLATNESS

Let $L(\theta)$ denote the training loss with respect to the model parameters $\theta \in \mathbb{R}^d$. A measure of local geometry around $\theta$ is the Hessian matrix $H(\theta) = \nabla^2 L(\theta)$. While individual eigenvalues of $H(\theta)$ provide direct information about sharpness in specific directions, computing the full spectrum is infeasible for modern networks. Instead, we consider the Frobenius norm of the Hessian:

$$C(\theta) := \operatorname{tr}\big(H(\theta)^2\big) = \sum_{j,k} \left(\frac{\partial^2 L(\theta)}{\partial \theta_j \partial \theta_k}\right)^2 = \|\nabla^2 L(\theta)\|_{\text{Fro}}^2. \qquad (1)$$

In particular, $C(\theta)$ aggregates the squared magnitudes of all second-order derivatives of the loss function. Intuitively, if $C(\theta)$ is small, then all entries of the Hessian are small, implying that the landscape around $\theta$ changes slowly. Thus, regions with low $C(\theta)$ can be regarded as *flat*, while large values of $C(\theta)$ correspond to sharp or rapidly varying regions of the loss surface. By penalizing $C(\theta)$ during training, we aim to bias the optimization process towards parameter regions that exhibit the desirable flatness.

## 2.2 Estimating Flatness using Hutchinson's Estimator

Direct computation of the Hessian $H(\theta)$ is infeasible due to the high dimensionality of the parameter space. To approximate the flatness measure $C(\theta) = \mathrm{tr}(H(\theta)^2)$, we adopt Hutchinson's stochastic trace estimator (Hutchinson, 1989) (See Algorithm 2). Hutchinson's stochastic trace estimator (Hutchinson, 1989), leverages the identity

$$\mathbb{E}_v\big[v^\top H(\theta)^2 v\big] = \mathrm{tr}\big(H(\theta)^2\big) = C(\theta), \tag{2}$$

for any random vector $v$ with zero mean and covariance matrix equal to the identity. This makes $v^\top H(\theta)^2 v$ an unbiased estimator of the flatness measure $C(\theta)$.

## 2.3 The Flatness Penalty

We modify the training objective by adding a flatness-aware regularization term. For a mini-batch $(x, y)$, the total loss is defined as:

$$L_{\text{total}} = L_{\text{task}}(f_\theta(x), y) + \lambda \cdot \tfrac{1}{B} \mathrm{tr}(H(\theta)^2), \tag{3}$$

where $L_{\text{task}}$ is the task-specific loss, $\lambda$ is the regularization coefficient, $B$ is the batch size (for scale normalization), and $\mathrm{tr}(H(\theta)^2)$ is estimated via Hutchinson's method (Algorithm 2).

Our hypothesis here is that including a flatness penalty term will encourage the optimization trajectories to prefer parameter regions where the Hessian entries are small, and create a bias towards flatter solutions, which is expected to improve generalization.

Our proposed training algorithm is as follows:

---

**Algorithm 1** Training with Flatness-Aware Regularization (FA-Regularization)

---

**Require:** Training data $\mathcal{D} = \{(x_i, y_i)\}_{i=1}^N$, model $f_\theta$, learning rate $\eta$, regularization coefficient $\lambda$, number of Hutchinson samples $M$, batch size $B$, number of epochs $T$
**Ensure:** Trained model parameters $\theta$
 1: Initialize model parameters: $\theta \leftarrow$ random initialization
 2: **for** $t = 1$ to $T$ **do**
 3:     Shuffle $\mathcal{D}$ and divide into mini-batches of size $B$
 4:     **for** each mini-batch $(x, y)$ **do**
 5:         $\hat{y} \leftarrow f_\theta(x)$
 6:         $L_{\text{task}} \leftarrow L_{\text{task}}(f_\theta(x), y)$
 7:         $g \leftarrow \nabla_\theta L_{\text{task}}$
 8:         $C \leftarrow 0$
 9:         **for** $j = 1$ to $M$ **do**
10:             Sample $v_j \sim \mathcal{N}(0, I)$ or Rademacher
11:             $h_j \leftarrow \nabla_\theta(g^\top v_j)$                                    ▷ Hessian-vector product
12:             $C \leftarrow C + \|h_j\|^2$
13:         **end for**
14:         $L_{\text{flat}} \leftarrow \frac{C}{M \cdot B}$
15:         $L_{\text{total}} \leftarrow L_{\text{task}} + \lambda \cdot L_{\text{flat}}$
16:         Update parameters: $\theta \leftarrow \theta - \eta \cdot \nabla_\theta L_{\text{total}}$
17:     **end for**
18:     **if** convergence criteria met **then**
19:         **break**
20:     **end if**
21: **end for**

---

# 3 Experiments and Results

## 3.1 Experiments

We evaluate the proposed flatness-aware regularization on three benchmark tasks spanning vision, text, and tabular domains. **CIFAR-100** is a 100-class image classification dataset ($32 \times 32$ color

images, 50k training samples). **IMDB Movie Reviews** is a binary sentiment classification task (25k training examples of text; in our experiments we use the 20 Newsgroups text dataset as a proxy for this task), and **Breast Cancer Wisconsin** is a binary classification task on a tabular biomedical dataset (∼450 samples with 30 features). We use the standard training/test splits for each dataset.

### 3.1.1 MODELS AND ARCHITECTURES.

For each task, we select a simple model architecture to facilitate efficient curvature computation. For CIFAR-100, we employ a two-layer multilayer perceptron (MLP) with 512 and 256 hidden units (ReLU activations and 30% dropout), followed by a softmax output layer. This choice (rather than a CNN) keeps the model relatively small for manageable Hessian calculations while still providing a nonlinear classifier for the image data. For the text classification task (IMDB/20 Newsgroups), we use a logistic regression model — a single-layer linear classifier — to represent a simple baseline for high-dimensional sparse text features. Finally, for the Breast Cancer tabular dataset, we use an MLP with the same two-hidden-layer architecture as above (512 and 256 units with ReLU and dropout). All models are implemented in JAX, which provides automatic differentiation for efficient gradient and Hessian-vector product computations.

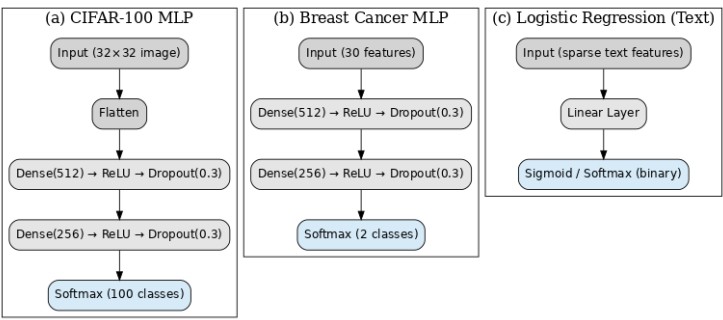

Figure 1: Flowchart of model architectures used in our experiments. The CIFAR-100 and Breast Cancer MLPs consist of input layers, dense layers with ReLU activations, dropout layers, and output layers (softmax for CIFAR-100, sigmoid for Breast Cancer). The text classification model uses a logistic regression architecture with a dense output layer and sigmoid activation.

### 3.1.2 TRAINING PROTOCOL.

Each model is trained using a cross-entropy loss and mini-batch optimization. We use the Adam optimizer for both MLP and logistic regression models (which we found to converge reliably for these architectures). We tailor the training schedule to each dataset-model pair: for the CIFAR-100 MLP, we train for a relatively short duration (e.g., 10 epochs with batch size 128) and use an initial learning rate of 0.01, decayed over time (a shorter schedule was sufficient given the limited capacity of the MLP on this vision task, and it helps constrain the overhead of second-order computations). The text (20 Newsgroups) logistic model and the Breast Cancer MLP are trained for up to 20–50 epochs (with smaller batch sizes, e.g. 32, and learning rates around 0.001–0.01) to ensure convergence. We apply a standard L2 weight decay to all models (e.g., $5 \times 10^{-4}$ for the CIFAR-100 MLP and $1 \times 10^{-4}$ for the others) to provide baseline regularization. To account for randomness in initialization and batch sampling, we repeat each training experiment for 3 independent runs and report the mean (and standard deviation where relevant) of the results. This provides statistical confidence in trends. We also conduct two-sample $t$-tests comparing each $\lambda > 0$ condition to the $\lambda = 0$ baseline to assess significance (following the evaluation protocol used in SAM).

### 3.1.3 FLATNESS-AWARE REGULARIZATION.

We integrate the proposed flatness-aware (FA) regularization into training via a curvature penalty on the loss landscape. The total loss for a mini-batch is defined as

$$L_{\text{total}} \;=\; L_{\text{task}} \;+\; \frac{\lambda}{B} \; \text{tr}(H^2) \,, \tag{4}$$

where $L_{\text{task}}$ is the standard cross-entropy loss on the batch, $H$ is the Hessian of $L_{\text{task}}$ with respect to the model parameters, and $B$ is the batch size. The term $\text{tr}(H^2)$ (the trace of the squared Hessian) serves as a measure of sharpness: a large value indicates a "sharp" minimum. We efficiently approximate $\text{tr}(H^2)$ via Hutchinson's stochastic trace estimator. In each training step, we sample a small number of random Rademacher vectors (e.g., 5 vectors) and compute their Hessian-vector products through automatic differentiation, using the identity $v^T H^2 v = \|Hv\|^2$. Averaging these quantities over the random vectors gives an unbiased estimate of $\text{tr}(H^2)$. This approach avoids constructing the full Hessian while still capturing the overall curvature. To reduce computational overhead, we calculate the curvature penalty at a lower frequency (for example, every 10 mini-batches) rather than at every update. During Hessian computations, we temporarily disable dropout and evaluate the loss on the current batch deterministically to obtain a stable curvature estimate. We experiment with several values of the regularization coefficient $\lambda$ to explore its effect: specifically, $\lambda \in \{0.0, 0.001, 0.01, 0.1, 1.0\}$. These choices range from no flatness penalty ($\lambda = 0$, our baseline) to very strong regularization ($\lambda = 1.0$). In between, 0.001 represents a light regularization, 0.01 a moderate setting expected to improve flatness with minimal impact on accuracy, and 0.1 a fairly strong regularizer.

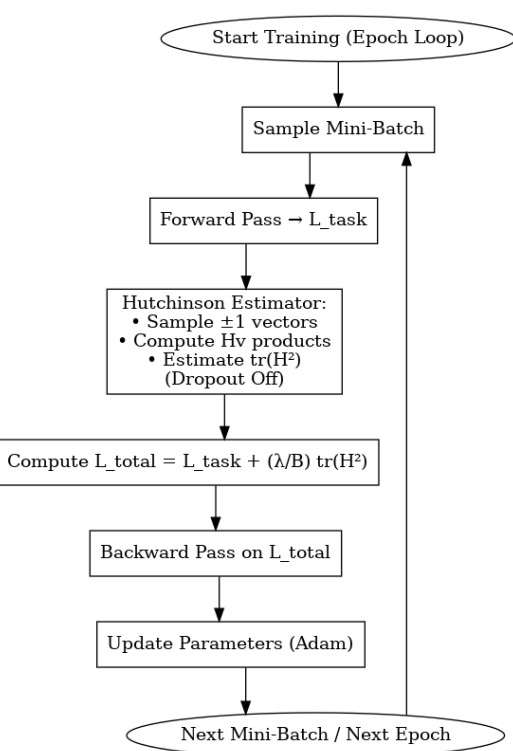

Figure 2: Flowchart of the training step with FA-regularization. For each mini-batch, a forward pass computes the standard task loss $L_{\text{task}}$. In parallel, the Hutchinson estimator draws random $\pm 1$ vectors and computes Hessian-vector products to estimate the curvature ($\text{tr}(H^2)$). The total loss $L_{\text{total}}$ is then obtained by adding the curvature penalty weighted by $\lambda$. Backpropagating $L_{\text{total}}$ updates the model parameters. This procedure encourages the optimizer to seek flatter minima by penalizing directions in parameter space with high curvature.

## 3.2 RESULTS

We conducted experiments on our three benchmark settings; image classification on CIFAR-100, text classification using a logistic model on IMDB (via 20 Newsgroups proxy), and binary classification on the Breast Cancer Wisconsin dataset—to evaluate how Flatness-Aware (FA) Regularization influences optimization dynamics, generalization, and computational cost. Across all tasks, we varied the flatness penalty $\lambda \in \{0, 0.001, 0.01, 0.1, 1.0\}$ and repeated each configuration for three runs to ensure stability. Here, $\lambda = 0$ is the training set up without our FA-regularizer.

Table 1: CIFAR-100 with FA regularization (MLP). The metrics we report here, are averaged across runs.

| $\lambda$ | Final Acc | Best Acc | Avg. Flatness | Train Time (s) |
|---|---|---|---|---|
| 0.000 | 0.2633 | 0.2800 | 0.000 | 60.85 |
| 0.001 | 0.2633 | 0.2833 | 23.05 | 2290.77 |
| 0.010 | 0.2700 | 0.2783 | 10.74 | 2433.53 |
| 0.100 | 0.2667 | 0.2967 | 4.14 | 2203.87 |
| 1.000 | 0.2583 | 0.2983 | 1.55 | 2337.86 |

Table 2: 20 Newsgroups logistic regression results (20 Newsgroups proxy). We report the values of the mean across 3 runs.

| $\lambda$ | Test Accuracy | Best Accuracy | Train Time (s) |
|---|---|---|---|
| 0.000 | $\sim$0.0535 | $\sim$ 0.0548 | 556.40 |
| 0.001 | $\sim$0.0535 | $\sim$ 0.0549 | 4917.94 |
| 0.010 | $\sim$0.0534 | $\sim$0.0548 | 4909.74 |
| 0.100 | $\sim$0.053 | $\sim$0.055 | 4900 |
| 1.000 | $\sim$0.053 | $\sim$0.055 | 4900 |

For evaluation, we adopted **test accuracy** as the primary measure of generalization instead of the conventional gap between training and test loss. This choice is motivated by the fact that for all $\lambda > 0$, flatness-aware (FA) regularization modifies the training objective by incorporating second-order information. As a result, the training loss no longer corresponds purely to the empirical risk but rather to a regularized surrogate. Using the train–test loss gap in this setting would therefore introduce bias, since the metric would partially reflect the regularization penalty rather than true generalization behavior. Accuracy, by contrast, is unaffected by the additional regularization term and provides a direct, unbiased measure of predictive performance on unseen data.

### 3.2.1 CIFAR-100 WITH MLP

We evaluated FA regularization on CIFAR-100 (Krizhevsky, 2009) using a 2-layer MLP with ReLU activations. Without regularization ($\lambda = 0$), the baseline achieved a final accuracy of $\sim 26.3\%$. We observed that introducing FA-regularization improved generalization at moderate strength; taining with the optimal $\lambda$ of 0.01 yielded $\sim 27.0\%$ final accuracy with reduced flatness, while larger $\lambda$ values decreased curvature but slightly degraded accuracy. In terms of efficiency, all FA runs were substantially slower ($\sim 2200$–$2400$ seconds) compared to the baseline ($\sim 60$ seconds) due to the second-order information used in the algorithm.

### 3.2.2 20 NEWSGROUPS SENTIMENT (LOGISTIC REGRESSION)

We used a logistic regression model on the 20 Newsgroups (Mitchell, 1997) dataset as a proxy for IMDB sentiment analysis. Across all $\lambda$ values, FA regularization neither improved nor degraded accuracy. However, it incurred significant computational overhead, increasing training time from $\sim 556$s (baseline) to nearly 4900s with FA.

### 3.2.3 BREAST CANCER (MLP)

On the Breast Cancer Wisconsin dataset (Wolberg & Street, 1993), FA regularization again showed mixed results. The baseline ($\lambda = 0$) averaged $55.6\%$ accuracy but with high variance ($\pm0.24$). Small regularization ($\lambda = 0.001, 0.01$) maintained similar average accuracy ($\sim 55.8\%$) with high variance, while training time increased more than tenfold ($\sim 160$s vs. 14s).

Table 3: Breast Cancer MLP results with FA regularization. We report the values of the mean across 3 runs.

| $\lambda$ | Test Accuracy | Best Accuracy | Train Time (s) |
|---|---|---|---|
| 0.000 | $\sim 0.5556$ | $\sim 0.5556$ | 14.36 |
| 0.001 | $\sim 0.5585$ | $\sim 0.5585$ | 162.06 |
| 0.010 | $\sim 0.5859$ | $\sim 0.5859$ | 166.00 |

## 4 RELATED WORKS

Understanding the geometry of the loss landscape has become central to explaining generalization in deep neural networks. Early work by Hochreiter & Schmidhuber (1997) hypothesized that flatter minima yield better generalization, while Keskar et al. (2017) showed that small-batch SGD tends to converge to flatter regions. This motivated algorithms like Entropy-SGD (Chaudhari et al., 2017), which bias training away from sharp basins, and visualization methods such as "filter normalization" by Li et al. (2018), which linked architectural choices to loss geometry. More recent approaches directly incorporate flatness into optimization objectives, most notably Sharpness-Aware Minimization (SAM) (Foret et al., 2021), which minimizes the worst-case loss within a neighborhood around the parameters. SAM and its many variants (Wu et al., 2023; Behdin et al., 2022; Du et al., 2022) demonstrate consistent generalization gains, though often at increased computational cost or with trade-offs in stability. At the same time, theoretical critiques (Dinh et al., 2017) caution that flatness is not always invariant under reparametrization, suggesting the need for more principled approaches.

Building on this line of work, our proposed Flatness-Aware (FA) Optimization explicitly leverages curvature information to guide training toward flatter regions of the loss landscape. Unlike SAM, which perturbs gradients to indirectly avoid sharp minima, FA-regularization directly incorporates second-order cues in a lightweight manner using the Hutchinson Estimator. This flatness-aware formulation provides both a more stable pathway to flat minima and a distinct mechanism from prior proposals.

## 5 CONCLUSION

In this work, we introduced Flatness-Aware Regularization (FA-Regularization), a simple yet principled approach that explicitly penalizes curvature in the loss surface using an efficient Hutchinson-based estimator of the Hessian trace. Our experiments across vision (CIFAR-100), text (20 Newsgroups/IMDB proxy), and tabular (Breast Cancer) domains demonstrate that FA-Regularization can guide optimization toward flatter minima and, in some cases, yield improved generalization. In particular, on CIFAR-100 with an MLP classifier, moderate regularization (e.g., = 0.01) consistently improved test accuracy relative to the baseline, lending empirical support to the hypothesis that flat minima can promote better generalization.

At the same time, our results highlight important trade-offs and limitations. First, FA-Regularization introduces significant computational overhead, with training times often increasing by an order of magnitude compared to baseline optimization. This suggests that while curvature-aware penalties are theoretically appealing, their practicality in large-scale deep learning may be limited unless more efficient approximations are developed. Second, the benefits of FA-Regularization were task- and model-dependent: while modest improvements were observed on CIFAR-100, no clear gains emerged in text or tabular classification tasks, where regularization mainly increased computational cost without improving accuracy. These findings align with broader debates in the literature, where flatness is shown to correlate with, but not fully explain, generalization.

Discussion

Our study provides further evidence that explicitly enforcing flatness can help in certain regimes but is not a universal solution to the generalization problem. In particular, the sensitivity to the regularization strength suggests that curvature-based regularization introduces a bias–variance trade-off:

too little regularization fails to enforce flatness, while too much can over-constrain optimization and harm predictive performance. Furthermore, the limited impact observed on non-vision datasets indicates that flatness alone may not capture the dominant generalization mechanisms in all domains. This underscores the need for a more nuanced understanding of when flatness is beneficial and how it interacts with dataset complexity, model architecture, and optimization dynamics.

Several promising directions arise from this work:

Scaling FA-Regularization – Our current implementation incurs substantial overhead due to repeated Hessian-vector computations. Future work could explore variance-reduced estimators, sub-sampling strategies, or low-rank Hessian approximations to make curvature penalties feasible for large-scale models such as CNNs or Transformers.

Hybrid Regularization – Flatness-aware penalties could be combined with other regularizers (e.g., weight decay, dropout, adversarial training, or SAM) to exploit complementary strengths. Studying such combinations may yield more consistent gains across modalities.

Architecture-Dependent Effects – Our results suggest that FA-Regularization has a stronger impact on vision tasks than on text or tabular tasks. A deeper investigation into the interaction between loss landscape geometry and architecture types (e.g., CNNs, RNNs, Transformers) could clarify when flatness-based methods are most effective.

Theoretical Characterization – While flatness is often associated with generalization, its formal role remains contested. Extending our framework with reparametrization-invariant flatness measures or connections to PAC-Bayesian generalization bounds could provide a stronger theoretical foundation.

Dynamic or Adaptive Regularization – Rather than fixing $\lambda$ throughout training, future approaches could adaptively adjust the flatness penalty based on training dynamics or curvature spikes, potentially reducing overhead while maintaining generalization benefits.

In summary, FA-Regularization contributes to the ongoing discourse on the relationship between flat minima and generalization. Our findings suggest that while explicit flatness penalties can improve performance in certain regimes, their effectiveness is highly context-dependent and computationally costly. We hope this work stimulates further exploration of scalable, principled methods for integrating curvature awareness into optimization, ultimately advancing our understanding of robust generalization in deep learning.

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

## A    APPENDIX

---

**Algorithm 2** Hutchinson-based Estimation of Flatness

---

1: Compute the first-order gradient: $g \leftarrow \nabla_\theta L$
2: Sample a random vector $v \sim \mathcal{N}(0, I)$ or from a Rademacher distribution
3: Compute the Hessian-vector product via automatic differentiation: $h \leftarrow Hv = \nabla_\theta(g^\top v)$
4: Form the estimate of flatness: $C(\theta) \leftarrow v^\top H^2 v = \|h\|^2$

---

