```python
import jax
import jax.numpy as jnp
from jax import grad, random
from typing import Dict, Callable

def cross_entropy_loss(logits: jnp.ndarray, labels: jnp.ndarray) ->
jnp.ndarray:
    """Compute cross-entropy loss"""
    num_classes = logits.shape[-1]
    y_onehot = jax.nn.one_hot(labels, num_classes)
    log_probs = jax.nn.log_softmax(logits)
    return -jnp.mean(jnp.sum(y_onehot * log_probs, axis=-1))

def compute_accuracy(logits: jnp.ndarray, labels: jnp.ndarray) ->
jnp.ndarray:
    """Compute classification accuracy"""
    predictions = jnp.argmax(logits, axis=-1)
    return jnp.mean(predictions == labels)

def hutchinson_trace_estimator(params: Dict, loss_fn: Callable, x:
jnp.ndarray,
                               y: jnp.ndarray, num_samples: int,
                               key: jax.random.PRNGKey) -> jnp.ndarray:
    """
    Hutchinson trace estimator for computing tr(H^2) where H is the
Hessian.
    This estimates the "flatness" of the loss landscape.
    """

    # Get gradient function
    grad_fn = grad(loss_fn, argnums=0)

    # Compute initial gradient
    g = grad_fn(params, x, y)
    g_flat, unflatten_fn = jax.flatten_util.ravel_pytree(g)
```

```python
    def hvp_fn(v: jnp.ndarray) -> jnp.ndarray:
        """Hessian-vector product using forward-mode autodiff"""
        def g_dot_v(p):
            g_params = grad_fn(p, x, y)
            g_flat, _ = jax.flatten_util.ravel_pytree(g_params)
            return jnp.dot(g_flat, v)

        hvp = grad(g_dot_v)(params)
        hvp_flat, _ = jax.flatten_util.ravel_pytree(hvp)
        return hvp_flat

    # Sample random vectors and compute trace estimate
    keys = random.split(key, num_samples)
    total_hvp_norm_sq = 0.0

    for i in range(num_samples):
        # Sample random Rademacher vector
        v = random.rademacher(keys[i], g_flat.shape, dtype=g_flat.dtype)

        # Compute H*v
        hvp = hvp_fn(v)

        # Accumulate ||H*v||^2 = v^T H^T H v = v^T H^2 v (since H is
symmetric)
        total_hvp_norm_sq += jnp.sum(hvp ** 2)

    return total_hvp_norm_sq / num_samples

def ca_regularization_loss(params: Dict, forward_fn: Callable, x:
jnp.ndarray,
                           y: jnp.ndarray, lambda_reg: float,
num_hutchinson_samples: int,
                           key: jax.random.PRNGKey, training: bool = True)
-> jnp.ndarray:
    """
    Curvature-Aware (CA) regularization loss function.

    Total loss = Cross-entropy loss + λ * tr(H^2)
    where H is the Hessian of the cross-entropy loss.
```

```python
    """

    # Task loss (cross-entropy)
    logits = forward_fn(params, x, training=training, key=key)
    task_loss = cross_entropy_loss(logits, y)

    if lambda_reg == 0.0:
        return task_loss

    # Define loss function for Hessian computation
    def task_loss_fn(p, xb, yb):
        logits_inner = forward_fn(p, xb, training=False)  # No dropout for
Hessian
        return cross_entropy_loss(logits_inner, yb)

    # Compute curvature penalty using Hutchinson estimator
    curvature_penalty = hutchinson_trace_estimator(
        params, task_loss_fn, x, y, num_hutchinson_samples, key
    )

    # Normalize by batch size for stability
    batch_size = x.shape[0]
    curvature_penalty = curvature_penalty / batch_size

    # Total loss
    total_loss = task_loss + lambda_reg * curvature_penalty

    return total_loss

def standard_loss(params: Dict, forward_fn: Callable, x: jnp.ndarray,
                  y: jnp.ndarray, weight_decay: float = 0.0,
                  key: jax.random.PRNGKey = None, training: bool = True) ->
jnp.ndarray:
    """Standard cross-entropy loss with optional L2 regularization"""

    # Task loss
    logits = forward_fn(params, x, training=training, key=key)
    task_loss = cross_entropy_loss(logits, y)
```

```python
    # L2 weight decay
    if weight_decay > 0.0:
        l2_penalty = 0.0
        param_count = 0

        for param_name, param_value in params.items():
            if 'weights' in param_name:  # Only regularize weights, not
biases
                l2_penalty += jnp.sum(param_value ** 2)
                param_count += param_value.size

        # Normalize by parameter count
        if param_count > 0:
            l2_penalty = l2_penalty / param_count
            task_loss += weight_decay * l2_penalty

    return task_loss

def create_loss_fn(loss_type: str, lambda_reg: float = 0.0, weight_decay:
float = 0.0,
                   num_hutchinson_samples: int = 5):
    """Factory function to create loss functions"""

    if loss_type == 'ca_regularized':
        def loss_fn(params, forward_fn, x, y, key=None, training=True):
            return ca_regularization_loss(
                params, forward_fn, x, y, lambda_reg,
                num_hutchinson_samples, key, training
            )
        return loss_fn

    elif loss_type == 'standard':
        def loss_fn(params, forward_fn, x, y, key=None, training=True):
            return standard_loss(params, forward_fn, x, y, weight_decay,
key, training)
        return loss_fn

    else:
        raise ValueError(f"Unknown loss type: {loss_type}")
```

```python
def evaluate_model(params: Dict, forward_fn: Callable, X_test: jnp.ndarray,
                   y_test: jnp.ndarray, batch_size: int = 1000) -> Dict:
    """Evaluate model on test set"""

    n_samples = X_test.shape[0]
    n_batches = (n_samples + batch_size - 1) // batch_size

    total_loss = 0.0
    total_accuracy = 0.0

    for i in range(n_batches):
        start_idx = i * batch_size
        end_idx = min((i + 1) * batch_size, n_samples)

        X_batch = X_test[start_idx:end_idx]
        y_batch = y_test[start_idx:end_idx]

        # Forward pass (no training mode, no dropout)
        logits = forward_fn(params, X_batch, training=False)

        # Compute metrics
        batch_loss = cross_entropy_loss(logits, y_batch)
        batch_accuracy = compute_accuracy(logits, y_batch)

        batch_weight = (end_idx - start_idx) / n_samples
        total_loss += batch_loss * batch_weight
        total_accuracy += batch_accuracy * batch_weight

    return {
        'loss': float(total_loss),
        'accuracy': float(total_accuracy)
    }
```