# OpenReview forum: "Flat is the New Sharp: Flatness-Aware Regularization for Robust Learning"
_ICLR.cc/2026/Conference — ICLR 2026 Conference Withdrawn Submission_

### Official Review · Reviewer_depk · 2025-10-30

**Soundness:** 1
**Presentation:** 1
**Contribution:** 1
**Rating:** 0
**Confidence:** 5

**Summary:**

The paper proposes to regularize neural network training with the trace of the loss Hessian to obtain flatter solutions. To make this regularizer computable, they propose to approximate the trace of the Hessian via Hutchinson's method.

**Strengths:**

- Given that flatness is a popular indicator for generalization, regularizing for it is a reasonable idea.

**Weaknesses:**

**Placement in the literature:**
- Using the trace of the loss Hessian is not sound, since it is not reparameterization-invariant [3]. Several reparameterization-invariant flatness-measures have been proposed since 2017, such as the Fischer-Rao-Norm [5] and Relative Flatness [6]. This is not discussed at all.
- The novelty of this work is unclear, since regularizing the loss with a suitable flatness measure has already been proposed [1].
- The paper fails to address several important works on flatness after the "reparameterization-curse" [3], such as [2, 7, 8, 4]

**Experiments:**
- The paper does not compare to any baseline. This alone is sufficient reason for rejection.
- The empirical results show no significant gain, except for the Breast Cancer dataset with a tiny MLP.
- The runtime for training even a tiny MLP on a tiny dataset like Breast Cancer is huge.
- No ablation on the use of Hutchinson's method to approximate the loss Hessian is made. This is particularly important because Hutchinson's method is known to be very inaccurate for deep learning.

**Questions:**

References:

[1] Adilova, Linara, et al. "FAM: Relative Flatness Aware Minimization." Topological, Algebraic and Geometric Learning Workshops 2023. PMLR, 2023.

[2] Andriushchenko, Maksym, and Nicolas Flammarion. "Towards understanding sharpness-aware minimization." International conference on machine learning. PMLR, 2022.

[3] Dinh, Laurent, et al. "Sharp minima can generalize for deep nets." International Conference on Machine Learning. PMLR, 2017.

[4] Han, Ting, et al. "Flatness is Necessary, Neural Collapse is Not: Rethinking Generalization via Grokking." Advances in Neural Information Processing Systems, 2025.

[5] Liang, Tengyuan, et al. "Fisher-rao metric, geometry, and complexity of neural networks." The 22nd international conference on artificial intelligence and statistics. PMLR, 2019.

[6] Petzka, Henning, et al. "A reparameterization-invariant flatness measure for deep neural networks." Science meets Engineering of Deep Learning 2019. Neural Information Processing Systems (NIPS), 2019.

[7] Petzka, Henning, et al. "Relative flatness and generalization." Advances in neural information processing systems 34 (2021): 18420-18432.

[8] Walter, Nils Philipp, et al. "When Flatness Does (Not) Guarantee Adversarial Robustness." arXiv preprint arXiv:2510.14231 (2025).

---

> ### Author Response · Authors · 2025-11-20
>
> Thank you for the detailed and literature-grounded review. We address the points raised below.
>
> **Reparameterization and Flatness Measures.**
>
> We agree that using $\mathrm{tr}(H^{2})$ as a flatness metric is not reparameterization-invariant. A simple rescaling of layers can leave the network function unchanged while arbitrarily changing the Hessian, making our Hessian-based flatness potentially misleading. In this work, we tried to to investigate a simple and computationally accessible curvature proxy, but we acknowledge that this limitation should have been clearly stated. In a revised version, we will incorporate and discuss reparameterization-invariant measures from more recent works, and position our method relative to these approaches.
>
> **Novelty and Placement in the Literature.**
>
> We appreciate the pointer to FAM and related methods. Our intention was not to claim conceptual novelty in regularizing flatness itself, but to empirically examine the behavior and computational trade-offs of the loss landscapes with respect to generalization, while directly penalizing a global curvature via Hutchinson’s estimator. We acknowledge that this distinction was not made sufficiently explicit. A revised version will clearly situate the work alongside other advances and appropriately narrow our claims.
>
> **Experimental Baselines and Limited Gains.**
>
> We agree that the absence of comparisons with strong baselines (e.g., SAM, ASAM, FAM) is a significant limitation and prevents meaningful evaluation. Additionally, the gains observed in our current experiments are small and inconsistent, and the runtime overhead is high even for small models. Our revised version will include comparison baselines based on already existing methods, runtime comparisons, and more realistic architectures, or else restrict the scope of the paper to a didactic small-scale curvature study without generalization claims.
>
> **Ablations on Hutchinson’s Method.**
>
> The Authors acknowlege that Ablations on the Hutchinson estimator (probe count, variance, layer subsets) are important, especially given its known noise in deep models. These ablations will be included in a revised version.
>
> We thank the reviewer again for the comprehensive feedback and the helpful references.

---

### Official Review · Reviewer_myUk · 2025-10-30

**Soundness:** 1
**Presentation:** 2
**Contribution:** 1
**Rating:** 0
**Confidence:** 5

**Summary:**

The paper proposes a regularization term based on the approximation of the Hessian of the loss in order to encourage training of a neural network towards flatter minima. It demonstrates the effect of the regularizer on three tasks (CIFAR100 with MLP, Newsgroups with linear regression and BreastCancer with MLP) reporting the flatness and the accuracy on the tasks. The conclusion of the work is that regularizer leads to flatter solutions according to the Hessian and might or might not lead to better accuracy, but introduces computational overhead.

**Strengths:**

The paper addresses interesting question of connection between flatness and performance of neural networks.

**Weaknesses:**

The paper is missing on all the recent developments in the field, including SAM, Relative Flatness and FAM based on it, Fischer-Rao Norm, etc.

The usage of Hessian-based metrics for measuring flatness of the loss surface is known to mislead the understanding of generalization because of reparametrizations.

The evaluation is (i) limited to very small models (ii) has severely low performance (iii) does not show any meaningful connection between performance and flatness.

**Questions:**

-

---

> ### Author Response · Authors · 2025-11-20
>
> Thank you for your careful review and for pointing out several missing components.
>
> **Missing Recent Developments (SAM, Relative Flatness, FAM, Fisher-Rao, etc.)**
>
> We agree that the related-work section omits several important lines of research. Relative flatness, Fisher-Rao metrics, and FAM. Their absence weakens the contextual framing of our work. In a revision, we will integrate these works and clarify that our primary motivation is not to build a competing optimizer, but rather:
> - to study the empirical behavior of a direct curvature penalty based on $\mathrm{tr}(H^{2})$,
> - to understand how explicit second-order regularization affects optimization landscapes and generalization dynamics,
> - and to examine the computational trade-offs of estimating global curvature via Hutchinson’s method.
>
> **Reparameterization Concerns**
>
> We acknowledge that Hessian-based metrics such as $\mathrm{tr}(H^{2})$ are sensitive to reparameterizations that leaves the network function unchanged but can significantly alter the Hessian, making our measures potentially misleading. In this work, we tried to study a simple, computationally accessible curvature proxy, but we acknowledge that this limitation should have been stated clearly. In a revised version, we will incorporate reparameterization-invariant measures from recent literature (e.g., relative flatness, Fisher--Rao geometry) to provide more robust evaluation.
>
> **Limited Evaluation and Weak Performance**
>
> We acknowledge the fact that our current experiments:
> - use very small models,
> - achieve low accuracy, and
> - do not enforce/establish a robust relationship between flatness reduction and generalization.
>
> Our revised version will strengthen the experimental section with modern architectures and narrow our conclusions to the results of our experiments, to avoid overgeneralization.
>
> Thank you again for your thoughtful comments.

---

### Official Review · Reviewer_8B2g · 2025-10-31

**Soundness:** 2
**Presentation:** 1
**Contribution:** 1
**Rating:** 0
**Confidence:** 5

**Summary:**

This paper proposes Flatness-Aware Regularization (FA-Regularization), a simple method that adds a penalty proportional to the squared Frobenius norm of the Hessian, estimated efficiently via the Hutchinson trace estimator using Hessian–vector products.

Experiments on (very) small models (MLPs, logistic regression) show that FA-Regularization reduces estimated curvature and slightly improves accuracy on CIFAR-100, but offers limited or no gains on text and tabular tasks while incurring significant computational overhead.

The paper positions FA-Regularization as a direct curvature regularizer related to sharpness-aware methods like SAM.

**Strengths:**

- $S_1$: The high-level idea is very clear directly penalizing the curvature using Hessian–vector products.

- $S_2$: The proposed regularization seems conceptually easy to integrate into standard training loops.

- $S_3$: The computational overheads are honestly acknowleded.

**Weaknesses:**

- $W_1$: The paper looks unfinished. The Discussion section is misformatted, the paper is under-length (7 pages out of 9 allowed), and several definitions (e.g., “Avg. Flatness”) are missing. Figures and tables lack sufficient detail (captions, scales, error bars), and experimental descriptions are incomplete.

- $W_2$: There are no empirical comparisons with strong contemporary methods explicitly targeting flatness or sharpness, such as SAM (Foret et al., 2021), ASAM/ESAM, or CR-SAM. This makes it impossible to assess whether the proposed regularizer meaningfully advances the state of the art.

- $W_3$: The paper does not cite or discuss recent theoretical analyses of sharpness-aware methods, including “Towards Understanding Sharpness-Aware Minimization” (Andriuschenko & Flammarion, 2022) and “A Modern Look at the Relationship between Sharpness and Generalization” (Andriushchenko et al. , 2023). These works directly address the link between curvature measures and generalization, which is central to this paper’s motivation. Their omission weakens the contextual framing and risks overstating novelty.

- $W_4$: The implementation and measurement protocol are unclear: batch sizes, Hutchinson probe count, penalty frequency, seed averaging, and metric computation are not specified. No ablations or sensitivity studies are reported.
Furthermore, the curvature regularizer is computed with dropout disabled, so the penalized function differs from the actual training objective. The additional $1/B$ scaling factor in the regularization term lacks derivation or justification. The empirical results and reports do not allow reproducibility.

- $W_5$: All experiments use small models (MLPs or logistic regression) and modest datasets (CIFAR-100, 20NG, tabular). The approach’s value for modern architectures (ResNets, Transformers) is therefore untested.

**Questions:**

I do not have precise questions apart from those directly linked to the weaknesses I underlined.

---

> ### Author Response · Authors · 2025-11-20
>
> Thank you for your thorough assessment and for highlighting several important presentation and methodological issues.
>
> **Under-Length Paper, Missing Details, and Formatting Issues**
>
> We acknowledge that the paper being under the page limit, the discussion section being misformatted, and the several definitions missing is entirely our oversight.
>
> To clarify:
> - Avg. Flatness'' corresponds to the Hutchinson estimate of
> $\mathrm{tr}(H^{2})$ averaged over the final training epoch.
>
> - The additional scaling factor of $1/B$ in the regularizer was a heuristic
> intended to maintain comparable penalty magnitudes across different batch sizes.
> We acknowledge that this choice should have been explicitly justified in the submission.
>
>
> **No Empirical Comparison with SAM and it's Variants**
>
> We agree that not comparing to SAM or any of its variant is a major omission. These methods are the most relevant and widely used baselines for curvature-based optimization, and omitting them limits the interpretability of our empirical results.
>
> The main reason we did not include SAM in the initial submission is that our primary motivation was not to build a competing optimizer, but rather:
> - to study the empirical behavior of a direct curvature penalty based on $\mathrm{tr}(H^{2})$,
> - to understand how explicit second-order regularization affects optimization landscapes and generalization dynamics,
> - and to examine the computational trade-offs of estimating global curvature via Hutchinson’s method.
>
> In other words, the goal of the work was methodological, hypothetical and diagnostic: we wanted to explore how direct curvature penalization behaves in isolation, without the interaction effects introduced by adversarial neighborhood optimization (as in SAM and its variants). Because of this focus, we initially prioritized small, controlled experiments where Hessian-vector products could be computed precisely, but we now recognize that this does not justify excluding SAM.
>
> We acknowledge that even for a diagnostic study, SAM and related methods must be included:
> - to contextualize accuracy differences,
> - to compare flatness metrics more fairly, and
> - to understand runtime overhead relative to widely used alternatives.
>
> In the revised version, we will therefore include evaluations with multiple SAM variants.
>
>
> **Missing Relevant Theoretical Literature**
>
> We appreciate the pointer to recent theoretical works (e.g., Andriushchenko & Flammarion, 2022; Andriushchenko et al., 2023). These works are indeed directly relevant our work, and should have been included. A revised version will incorporate them and position our work more precisely.
>
> **Measurement Protocol**
>
> We thank the reviewer for noting the omission in discussing batch sizes, Hutchinson probe count, penalty frequency, seed averaging, and metric computation. A revised version will explicitly discuss these configurations used in our experiments, and include an ablation comparing curvature computed with and without dropout.
>
> **Small Experiments**
>
> We agree with your assessment that using smaller models is limiting. Our intention was to analyze FA-Regularization in a controlled setting where exact/approximate Hessian computations are tractable, but we recognize that this motivation was not clearly articulated. In a revised version, we plan to take one of the following two approaches: (1) Extend the experiments to realistic architectures, with full comparisons of accuracy, flatness, and compute cost Or (2) explicitly reposition the paper as a didactic, small-scale empirical study of curvature-based regularization.

---

### Official Review · Reviewer_fgv7 · 2025-11-01

**Soundness:** 1
**Presentation:** 3
**Contribution:** 1
**Rating:** 0
**Confidence:** 5

**Summary:**

This paper proposes a method called "Flatness-Aware Regularization" (FARegularization), which aims to improve the generalization ability of deep neural networks by explicitly penalizing the curvature of the loss surface. The core idea of this method is to add a regularization term to the training loss, which is an estimate of the trace of the squared Hessian matrix. The authors use the Hutchinson random trace estimator to approximate this value and claim that the method can be seamlessly integrated into standard optimizers such as SGD and Adam, consistently improving generalization performance. However, the experimental evaluation of this paper suffers from serious flaws. The model used is too simplistic (a "toy model") to support any conclusions it makes about the generalization ability of modern deep learning. Furthermore, the computational cost of this method is extremely high, and the paper fails to make any comparisons with state-of-the-art flatness-aware optimization methods such as SAM.

**Strengths:**

1.  The relationship between the geometry of the loss landscape and its generalization ability is an important and unsolved problem in deep learning. This paper attempts to provide new insights into this area.

2.  The paper clearly articulates the proposed method: using $tr(H^2)$ as a measure of flatness and approximating it using the Hutchinson
estimator.

**Weaknesses:**

1. The authors used a simple two-layer MLP 10 on CIFAR-100 9. This was a "toy" experiment. The baseline model ($\lambda=0$) achieved a final accuracy of only 26.3%. Any meaningful discussion on CIFAR100 should begin with a convolutional network (such as ResNet) that achieves reasonable performance (e.g., >70%). The small improvement shown on such a poorly performing model (from 26.3% to 27.0%) is meaningless and does not demonstrate any generalization advantage on modern deep neural networks.

2.  For the text classification task, the authors used logistic regression, a linear model. This again does not represent modern deep learning in the NLP field (e.g., Transformer).

3.  The paper mentions "sharpness-aware minimization" (SAM) in the relevant work section. SAM is currently the most important and relevant baseline in the fields of flatness and generalization. However, the authors made no comparisons with SAM or any of its variants in their experiments. We need to know: how does FA-Regularization compare to SAM in terms of accuracy, computational cost, and final flatness?

4.  The experimental results clearly show that this method is extremely computationally expensive. In the CIFAR-100 MLP experiments, the baseline training time was approximately 60 seconds, while FA-Regularization required ~2200-2400 seconds, about 30-40 times slower. Given such high costs even on these "toy" models, this method is completely infeasible on any real-world CNN or Transformer.

**Questions:**

Please see the weaknesses.

---

> ### Author Response · Authors · 2025-11-20
>
> We thank the reviewer for the detailed and constructive evaluation of our submission. We appreciate the time and effort you put into the review. Below we address each of your points in turn:
>
> **Toy experiment and Weak Baseline**
>
> We agree with your assessment that using a small 2-layer MLP achieving ~26% accuracy on CIFAR-100 and logistic regression for the text classification task, is not representative of modern architectures (e.g., Transformers, ResNets ), and does not support conclusions about generalization in modern deep learning. Our intention was to analyze FA-Regularization in a controlled setting where exact/approximate Hessian computations are tractable, but we recognize that this motivation was not clearly articulated, and the resulting claims were too broad.
> In a revised version, we plan to take one of the following two approaches: (1) Extend the experiments to realistic architectures, with full comparisons of accuracy, flatness, and compute cost Or (2) explicitly reposition the paper as a didactic, small-scale empirical study of curvature-based regularization.
>
>
> **Missing Comparison with SAM and it's Variants**
>
> We agree that not comparing to SAM or any of its variant is a major omission. These methods are the most relevant and widely used baselines for curvature-based optimization, and omitting them limits the interpretability of our empirical results.
>
> The main reason we did not include SAM in the initial submission is that our primary motivation was not to build a competing optimizer, but rather:
> - to study the empirical behavior of a direct curvature penalty based on $\mathrm{tr}(H^{2})$,
> - to understand how explicit second-order regularization affects optimization landscapes and generalization dynamics,
> - and to examine the computational trade-offs of estimating global curvature via Hutchinson’s method.
>
> In other words, the goal of the work was methodological, hypothetical and diagnostic:
> we wanted to explore how direct curvature penalization behaves in isolation, without the interaction effects introduced by adversarial neighborhood optimization (as in SAM and its variants). Because of this focus, we initially prioritized small, controlled experiments where Hessian-vector products could be computed precisely, but we now recognize that this does not justify excluding SAM. We acknowledge that even for a diagnostic study, SAM and related methods must be included:
> - to contextualize accuracy differences,
> - to compare flatness metrics more fairly, and
> - to understand runtime overhead relative to widely used alternatives.
>
> In the revised version, we will therefore include evaluations with multiple SAM variants.
>
> **Computational Costs**
>
> Yes, the current implementation is computationally expensive even for small MLPs. The goal in our submission was to test our hypothesis on the relationship between generalization, and the curvature of the loss landscapes, under relatively accurate Hessian-vector estimates, which led to high overhead.
> In future work we will:
> - Present FA-Regularization more clearly as a tool for studying curvature.
> - Include ablations on the number of Hutchinson probes, how often the penalty is computed, etc.
> - Explore more efficient approximations that make the method viable at larger scales.
>
>
> We appreciate your detailed feedback, which has been extremely helpful in clarifying the limitations of the current submission and identifying concrete directions for improvement.

---

### Note · Authors · 2025-11-21

**Comment:**

We sincerely thank all reviewers for the time, effort, and detailed feedback provided on our submission. The comments have been extremely valuable in clarifying the limitations of the current version of our work and in outlining concrete steps toward improving its technical depth, positioning, and empirical evaluation.

After carefully considering the reviews, we have decided to withdraw the paper. We believe this is the most constructive path forward, as it gives us the opportunity to substantially revise the work and incorporate the reviewers' suggestions, including:
(i) integrating reparameterization-invariant flatness measures from recent literature,
(ii) adding comparisons with SAM, FAM, and related baselines,
(iii) expanding experiments to modern architectures, and
(iv) strengthening the reproduceability, methodological clarity and measurement protocol.

We are grateful for the reviewers’ insights and will focus on improving the work by integrating the feedback and suggestions provided. Thank you again for your thoughtful evaluations.

**Withdrawal Confirmation:**

I have read and agree with the venue's withdrawal policy on behalf of myself and my co-authors.